# Automatic Determination of Secondary Dendrite Arm Spacing in AlSi-Cast Microstructures

**DOI:** 10.3390/ma14112827

**Published:** 2021-05-25

**Authors:** Christian Gawert, Rüdiger Bähr

**Affiliations:** Institute of Manufacturing Technology and Quality Management, Otto-von-Guericke-University Magdeburg, Universitätsplatz 2, 39106 Magdeburg, Germany; ruediger.baehr@ovgu.de

**Keywords:** SDAS, microstructure, aluminium alloys, image processing

## Abstract

A new procedure for the automatic measurement of the secondary dendrite arm spacing (SDAS) from microscopic images is presented. The individual primary and secondary dendrite arms are identified through suitable segmentation techniques and clustered in such a way that dendritic structures are obtained suitable for SDAS measurement. The algorithms are applied to two different hypoeutectic aluminum cast alloys, and the quality of the measurements obtained is assessed through a comparison to manually measured SDAS values. A good agreement between the automated measurements and the distribution of manual measurements is found for both cast structures considered. In addition, a decrease in computation time is observed which allows for an increase in measurement density that is used to characterize the microstructures.

## 1. Introduction

The significance of lightweight engineering increases continuously, particularly in the fields of automotive and rail vehicle industry. Reducing vehicle mass is becoming an increasingly important factor, especially for hybrid- and electro vehicles, to increase their range. Hence, the market for aluminum casting alloys is growing. In particular, AlSi alloys are used for automotive cast products because of their good castabillity.

Hypoeutectic AlSi cast alloys solidify usually in dendritic structures. The resulting structures of the grown dendrites are mainly influenced by the cooling rate of the solidification process [1,2,3,4,5]. In order to quantify those dendritic structures, the secondary dendrite arm spacing (SDAS) is commonly used. This SDAS value is known to correlate to a variety of different solid state properties of the cast, as for example the solid solution and age hardening, as well as the shrinkage or gas porosity. A reduction in hot tearing tendency and a better castability were observed at low SDAS values [6,7], and a decrease in fatigue life with increasing SDAS values is reported [8]. Also, the corrosion properties were shown to be related to the SDAS [6,9,10].

The SDAS quantification is performed either manually or semi-automatically by means of a prepared micrograph of the microstructure. The determination of the SDAS requires an expert operator to examine each field of interest to quantify the measurand. The operator has to draw a line through the elements under investigation, which is very time-consuming. As a consequence, mostly only a few sections can be investigated. Hence, it is desirable to automate the measurement of the SDAS to obtain an overall view of the microstructure with appropriate expense [11].

Despite this demand for algorithms to automatically measure the SDAS, only a few examples of successful realizations can be found in the current literature. It has been suggested to measure the dendrite cell size (DCS) on the basis of a circular method [12]. A DCS estimate is obtained in this approach by determining the dendrite-eutectic intersections along a set of given circles on the micrograph that is to be measured. However, the applicability of this approach to non-uniform structures has been questioned [13]. Instead, the authors suggest measuring the SDAS on the basis of a spacing transform, which allows for obtaining appropriate measurements within a 2D slice originating from a 3D microstructure. Both approaches avoid the explicit identification of dendritic structures that are suitable for SDAS measurements. This necessitates a post-processing step for both approaches, in which either the measured DSC value are regressed to manually measured SDAS values with a suitable approach [12] or in which only a certain part of the spacing distribution obtained is considered for SDAS measurement [13].

The algorithms of this work aim to complement the previous approaches in this respect. We present a method by which the individual dendrite arms can be detected, scored and grouped in such a way that dendritic structures can be identified that are suitable for the direct SDAS measurement. The algorithms developed for this purpose are presented and illustrated in Section 2 of this manuscript along with a procedure for the binarization of the micrograph. The resulting SDAS measurements are presented in Section 3 for the example of two qualitatively different cast structures. The results are assessed and discussed on the basis of a set of manual measurements that were obtained for the same microstructures. Finally, Section 4 concludes this article.

## 2. Materials and Methods

This section aims at introducing and illustrating the algorithms that are developed and employed in this work for measuring the secondary dendrite arm spacing. The algorithms are sub-divided into three major parts, namely: binarization, object segmentation and object clustering, which are presented in Section 2.1, Section 2.2 and Section 2.3. The individual processing steps that are presented in these sections require typically a set of parameters, which are compiled in Table 1, given at the end of Section 2.

### 2.1. Binarization

The aim of a binarization procedure is to identify all objects of interest in a given image and to separate them from the image background. This is equivalent to providing a logical (or binary) image mask in which all pixels belonging to the objects of interest are labeled as true, whereas all background pixels are labeled as false. Developing such a binarization procedure is particularly challenging in the context of identifying dendritic structures from microscopic images. This is because these dendrites appear typically as unstructured objects whose grayscale level deviate only slightly from the grayscale values of the eutectic background. Furthermore, the average grayscale level of the background is not constant as low-frequency noise arises in horizontal and vertical direction from a non-homogeneous illumination of the image. Hence, binarization through simple global thresholding is not feasible in this application. Instead, a four-step (steps (a) to (d)) binarization procedure is developed and illustrated in Figure 1.

All pixels belonging to the cast structure are identified in a preliminary processing step through a global thresholding procedure. The threshold that is required for this operation can be determined simply through Otsu´s algorithm [14] as the grayscale levels of all metal-pixels differ substantially from the dark image background. As illustrated in Figure 1b, essentially all metal-pixels are correctly labeled as true and, hence, the resulting binary image is used without any further changes in the remainder in this algorithm.

As stated earlier, low-frequency noise in the grayscale levels of the background pixels prohibit the application of a global thresholding procedure to identify the dendrite structures. To remedy this issue, the image is divided into a set of 10 × 10 sub-images in order to reduce the influence of the noise on the individual sub-images. An average grayscale value of these sub-images can be obtained on the basis of the intensity values of the metal pixels labeled as true in step a) of Figure 1. As the grayscale levels of the dendrite pixels are typically higher than the grayscale values of the background, the identification of the dendrite pixels can be achieved by the help of the average intensity level. In this work, all pixels satisfying the condition:(1)$$I_{x,y}\;\geq \;I_{mean}+T_{1}$$
are considered as background pixels. In this condition, $I_{mean}$ denotes the average grayscale value whereas $T_{1}$ is a threshold value, which is set to five in this work. The application of Equation (1) leads to binary images as shown in Figure 1c. It can be seen that this procedure leads to rather good results as most of the pixels are labeled correctly. However, some smaller artifact in both, the background as well as in the dendrite regions can be identified in which the binarization leads to false results. This issue is resolved in a subsequent filtering step described in the following paragraph.

In order to correct for the false artifacts resulting from the procedure of Equation (1), the initial binary image is smoothened by a Gaussian filter with an averaging mask of 5 × 5 pixels and a standard deviation of 5. This operation serves to reduce the intensity values of the falsely labeled pixels in the background region, whereas the intensity values of the pixels that belong to the dendrites but were labeled as false are increased (see Figure 1d). On the basis of this filtered image, a second thresholding is performed by labeling all pixels as either true or false depending on the fulfillment of the condition:(2)$$I_{x,y}\;\geq \;T_{2}$$

The threshold value $T_{2}$ that appears in this condition is set to 0.6, which leads to binary images as shown in Figure 1e. The number of falsely labeled pixels is significantly reduced (compared to the results of Figure 1c) by this approach and hence, this binary image is used as a basis for the subsequent processing steps.

As can be seen from the exemplary micrograph of the AlSi10Cu alloy shown in Figure 1, the dendrite density is rather high and only few pixels do actually belong to the eutectic background. As these background pixels separate the individual dendrites and dendrite arms, the distinguishability of the individual dendrite arms is hindered by this high dendrite density. In order to improve this situation, a morphological erosion operation with a disk-shaped structuring element with a size of 3 pixels is applied to the binary image resulting from Equation (2). Morphological erosion generally removes the outmost pixels of an object which conversely leads to an enlargement of the areas considered as background in a binary image. In the case of Figure 1, this enlargement of the background region leads to an improved separation between the individual dendrite arms. Therefore, the resulting binary image is used as input for the subsequent object segmentation and clustering steps of this algorithm, which are described in the next sections of this work.

### 2.2. Object Segmentation

Once a binary image is obtained from the procedure described above, all connected components of this image are identified on the basis of eight-connectivitiy, and all objects with an area above 50 pixels are passed to the subsequent processing steps. The objects that are extracted from the binary image are at this point in general not individual secondary dendrite arms. Instead, more complex structures consisting of several connected primary and secondary dendrite branches, as for example shown in Figure 2, have to be expected. Hence, it is necessary to decompose these complex objects in such a manner that the individual secondary dendrite arms can be identified from the individual connected components. For this purpose, a distance transform is applied on the inverted image (shown in Figure 2b) of the connected component. The image obtained through this distance transform (illustrated in Figure 2c) is afterwards passed to a watershed transform which segments the entire object into smaller, essentially convex, regions (see Figure 2d).

In this procedure, a candidate region i is selected, and all neighboring regions j are considered for the merging procedure. Based on the observation that the individual dendrite arms are objects that are approximately convex, the solidity $S_{i,j}$ is defined as the ration between the area of both regions and the area of the joint convex hull of the candidate–neighbor combination:(3)$$S_{i,j}=\frac{A_{i}+A_{j}}{A_{CH,i,j}}$$

This is calculated for all candidate-neighbor combinations. The neighbor region with the maximal solidity value is the merged with the candidate region provided that the solidity value exceeds a threshold value of S ≥ 0.85. In this case, the neighborhood relations of all regions are redefined, a new candidate is selected, and the neighbor selection procedure is re-initiated. If no candidate was found to fulfill the solidity threshold, the next candidate region is selected and all its neighbors are tested again. This iterative procedure is repeated until none of the remaining candidate regions has any neighbor which could be merged to this region. The result of this procedure is shown in Figure 2e. As can be seen, a significant reduction of the number of regions is achieved (compared to Figure 2d) and the individual dendrite arms can be identified as single segmented regions.

### 2.3. Object Grouping and Cluster Scoring

The algorithm presented so far allows for the recognition of dendritic structures as well as for the identification of individual primary and secondary dendrite arms. However, in order to measure the SDAS, the individual objects need to be clustered in an appropriate manner in order to facilitate a reliable SDAS measurement.

To this end, the centroid coordinates of all segmented objects are calculated and transferred into a binary mask. A dilation operation is afterwards performed on this centroid mask with a square-shapes structuring element having a size of 10 pixels. This dilation is performed in order to improve the performance of the subsequent clustering step, which is based on a Hough transform. This Hough transform detects straight lines in the centroid mask which intersect with a maximal number of pixels within the mask, as shown in Figure 3b. An initial clustering of the segmented objects is achieved by this approach by considering the set of all segmented objects that intersect with an individual line as one cluster. Since the only information employed for the objects clustering are the positions of the object centroids, it has to be expected that the majority of the detected clusters contain dendrite arms belonging to more than just one dendrite branch (see for example Figure 3c. Therefore, these clusters cannot be considered as suitable for measuring the SDAS. However, as the number of detected lines—and thus object clusters—is relatively large (about 4000 for typical image sizes of 1 megapixel), it can be safely assumed that a sufficiently large number of object clusters is in fact suitable for SDAS measurement. Therefore, the remainder of this algorithm is dedicated to the identification of these suitable object clusters from the set of all object clusters. This identification is based on the following requirements and observations:A suitable cluster should be composed of a minimal number of objects.The distributions of distances between the individual objects should be as homogeneous as possible for a suitable object cluster.The average distance between the individual objects should not exceed a maximal value.Secondary dendrite arms appear typically with an elliptical shapes and the orientation of the major axis lengths of an equivalent ellipse is preferably perpendicular to the primary dendrite branch–and hence to the detected line.The average ellipse orientation should be perpendicular to the orientation of the detected line.High aspect ratios (measured by the ratio of major and minor axis length) indicate well-segmented secondary dendrite arms. Consequently, clusters with objects having higher aspect ratios are considered as better suited for SDAS measurement.

The first criterion of this list is rather easy to check, and hence all clusters consisting of fewer objects that the minimal number N_min_ (set to 6 in this work) are not further considered for SDAS measurement. In addition to this constraint on the minimal number of segments, also a maximal number N_max_ is defined and set to 10. Consequently, all segment clusters that consist of 6 to 10 subsequent segments of a specific line are tested for their suitability for SDAS measurement in the further processing steps.

The later criteria (b) to (f) are not as well-defined as criterion (a). It is furthermore possible (and has to be expected) that some suitable object clusters fulfill one or more of these conditions only partially while they perfectly fulfill other criteria of this list. The decision whether an object cluster can be considered suitable for SDAS measurement is hence deemed unreliable if only one of these criteria is considered. Instead, we develop measures for the fulfillment of every criterion and derive a scoring value $S_{c}$ which serves to assess the suitability of an object cluster for SDAS measurement. This scoring value is chosen to be a linear combination of the individual fulfillment measures according to:(4)$$S_{c}=w_{b}T_{b}+w_{c}T_{c}+w_{d}T_{d}+w_{e}T_{e}+w_{f}T_{f}$$
where $w_{b}$ to $w_{f}$ are linear factors (specified in Table 1) weighting the influence of the individual fulfillment measures on the scoring value $S_{c}$. In order to derive the measures related to the distance distribution among the individual dendrite arms, the average intersection point $P=\left(P_{x},\;P_{y}\right)$ is defined as:(5)$$P_{x}=\frac{P_{a,x}+P_{b,x}}{2}$$
(6)$$P_{y}=\frac{P_{a,y}+P_{b,y}}{2}$$

Here, $P_{a}$ and $P_{b}$ denote the coordinates of the first and last pixel of the current segment that intersect with the line that is to be scored. With the definition of Equations (5) and (6), the distance d_i_ between two neighboring segments (index i and i+1, respectively) is obtained by:(7)$$d_{i}=\sqrt{{\left(P_{i,x}-P_{i+1,x}\right)}^{2}+{\left(P_{i,y}-P_{i+1,y}\right)}^{2}}$$

Calculating the distances for all segments of the considered cluster yields a distance distribution, which is described by its mean value $\mu_{d}$ and its standard deviation $\sigma_{d}$ in the remainder of this work. Using $\mu_{d}$ and $\sigma_{d}$, the fulfillment measures for conditions (b) and (c) are calculated by:(8)$$T_{b}=\frac{\sigma_{d}}{\mu_{d}}$$
and
(9)$$T_{c}=max\left\{0{,}_{\mu_{d}}-\mu_{d,max}\right\},$$
respectively.

In order to derive fulfillment measures for conditions (d) to (f), the segmented regions that are associated to the current line are approximated by ellipses having the same central moments as the segmented regions (see [15] for more details). These ellipses are quantified by their major axis lengths $A_{maj,i},$ their minor axis lengths $A_{min,i}$ and their orientations $\theta_{i}$. Using these values, the measure $T_{d}d$ is calculated by:(10)$$T_{d}=\overset{n_{Seg}}{\underset{i=1}{\sum }}\frac{A_{maj,i}}{A_{min,i}}\left(cos\Delta \theta_{i}+1\right).$$

Note that the orientation deviations $\Delta \theta_{i}$ are scaled with the individual aspect ratios to account for the fact that the accuracy—and thus the reliability—with which the orientation can be determined is dependent on the aspect ratio. Similarly to Equation (10), the average orientation of the segment cluster considered is accounted for by:(11)$$T_{e}=cos\left(\frac{1}{n_{Seg}}\overset{n_{Seg}}{\underset{i=1}{\sum }}\Delta \theta_{i}+1\right).$$

Finally, the mean aspect ratio of the segments is taken as a measure for the fulfillment of condition (f) according to:(12)$$T_{f}=\frac{1}{n_{Seg}}\overset{n_{Seg}}{\underset{i=1}{\sum }}\frac{A_{maj,i}}{A_{min,i}}.$$

Having defined the measures $T_{b}$ to $T_{f}$, the scoring value $S_{c}$ can be calculated for every segment cluster, or subset thereof with subsequent segments, that contain $N_{min}$ to $N_{max}$ segments. With the weight $w_{b}$ to $w_{f}$ having the same signs as in Table 1, low scoring values $S_{c}$ indicate segment clusters that are suitable for SDAS measurement whereas clusters with high scoring values are not suitable for this purpose. Hence, all considered segment clusters are ranked according to their scoring value.

Due to the comparably large number of object clusters that is processed through the methods described above, it is possible that segment clusters or parts of it are processed and scored multiple times. Therefore, a final check is performed that deletes all segment clusters which have at least 3 segments in common with a cluster having a lower scoring value than the cluster that is to be deleted. This procedure ensures that only unique clusters are considered for the results of the SDAS measurement, which is presented and discussed in the next section of this manuscript.

## 3. Results and Discussion

This section aims at assessing and discussing the performance of the routines for SDAS measurement presented in the previous section. For this purpose, the algorithm is applied to two different AlSi standard cast alloys, and the results are compared with manual measurements for both cases in Section 3.1 and Section 3.2. In addition to this comparison, the computational performance of the algorithm is also analyzed in Section 3.3.

In order to evaluate the algorithms of the previous section, we choose two different hypoeutectic AlSi cast alloys with different microstructures. Both differ mainly by the fraction of the eutectic solid solution. The first microstructure consists of an AlSi10Cu cast alloy and contains a relative low fraction of eutectic solid solution. The α-aluminum dendrite structure is significantly denser, as compared to the second microstructure of this work., consisting of an AlSi11Mg cast alloy. In further explanation they will described as “structure 1” and “structure 2”, and their chemical compositions are given in Table 2.

The aim of assessing the quality of the SDAS measurements obtained by the procedures presented above necessitates the availability of a reliable standard against which the results can be compared. This standard is obtained from manual measurements from six different experts for both cast structures used in this work. Each individual manual measurement consists of 10 dendritic structures which are used to obtain an average SDAS value for the image considered. Hence, a distribution of manually measured SDAS values is obtained when all six results are considered. The quality of the automated SDAS measurements is assessed on the basis of this distribution which is quantified by its mean value and its standard deviation. The results of this assessment are presented in Section 3.1 and Section 3.2, where both microstructures are considered.

### 3.1. Secondary Dendrite Arm Spacing (SDAS) Measurements of Structure 1

The micrograph of structure 1 considered in this work has a size of 15,177 × 11,783 pixels, depicting a size of 14.72 mm × 11.43 mm. In order to obtain SDAS measurements of the core as well as of the exterior of this structure, the entire image is subdivided into 6 × 6 sub-images, and the diagonal of this array of sub-images is further analyzed. The dendrite structures measured for one of these sub-images are exemplarily shown in Figure 4 depicting one manual as well as the automated measurements. Obviously, the selection of dendrite structures is not identical between both measurements (as this is typically also the case among the individual manual measurements). This can in part be attributed to the large image size that results in a rather high number of dendrites within this image. In fact, most of the automatically selected structures are indeed dendrites that can be used for the determination of the SDAS. However, structure choices can also be found that would not have been chosen by a human expert. Hence, the quality of the SDAS measurements obtained needs to be assessed in a more quantitative manner.

The results of this assessment are given in Figure 5, where the performance of the automated SDAS measurements is compared to all results that were obtained manually. The left panel of this figure illustrates the distributions of the manually measured SDAS values. The mean values of these distributions are indicated by a thick solid line, and the regions spanned by the corresponding standard deviations are indicated by the shaded area. Two general trends can be seen from this sub-figure. The distributions of the manual measurements are rather broad, having a standard deviation of about two to three pixels, which corresponds roughly to 10% of the average SDAS values. This might be seen as an indication that a manual SDAS measurement is somewhat influenced by the dendrite selection of the human expert. This observation is even more apparent in the right panel of Figure 5, where the average SDAS measurements of the individual experts are shown as thin colored lines. Apart from the deviations among these measurements, already indicated in the left sub-figure, it can be observed that the SDAS measurements of most of the human experts are consistently either higher or lower than the average SDAS value. Also, intersections among the individual experts are comparably rare. This might be interpreted as an indication for a certain bias of the measured SDAS values, which is induced by the intuition of the individual human experts.

The automated SDAS measurements are shown as thick dashed lines in both subfigures. It is apparent, that some deviations to the average manual SDAS values occur. Nevertheless, these deviations are about the same order of magnitude as the standard deviation of the distribution of the manual measurements. In fact, only the automated SDAS measurement of the first sub-image is slightly outside of the region spanned by these standard deviations, while all other SDAS values lie within this region. From a statistical point of view, this observation indicates that the hypothesis that the automated SDAS measurements differ significantly from the manual ones needs to be rejected. This conclusion appears also to hold if systematic deviations between automated and manual measurements are considered, as no significant bias towards higher of lower SDAS values is apparent from Figure 5.

The statistical significance of the test described above could clearly be further improved by considering additional manual measurements to provide better estimates for the mean SDAS value and the standard deviations of the SDAS measurement distribution. However, more importantly, it needs to be checked whether the example of structure 1 considered here constitutes an ideal case in which the automated SDAS measurement procedure lead to satisfactory results. In order to elucidate this question, the next section considers an AlSi cast structure, denoted as structure 2, which differ significantly in terms of shape, size and density of the individual dendrite arms from structure 1 studied so far.

### 3.2. SDAS Measurements of Structure 2

The assessment of the measurements of structure 2 is similar as compared to the Section 3.1. The micrograph of the second microstructure used in this work, denoted by structure 2, has a size of 4328 × 9465 pixels, or 4.2 mm × 9.18 mm. Since the depicting size of structure 2 is smaller as compared to structure 1, this image is sub-divided into 4 × 4 sub-images in order to obtain images sizes comparable to structure 1. As in the case of Section 3.1, one diagonal of this set of sub-images is used for the further testing of the algorithms. A selection of the SDAS measurements for one of these sub-images is shown in Figure 6, depicting one set of manual as well as the automated measurements. As in the case of structure 1, there are dendritic structures, identified automatically from the algorithm, that would not have been chosen by a human expert.

By means of a comparison with the mean value of the manual measurements, the automated measurement is evaluated regarding the reproducibility. The results of this evaluation for structure 2 are shown in Figure 7. The comparison between the automated SDAS measurements and the manually measurements of the six experts, is performed similar to the evaluation of structure 1. The distributions of the manual measurements having a standard deviation of about two pixels, which corresponds roughly to 12% of the average SDAS values. The SDAS measurements of most of the human experts are consistently either higher or lower than the average SDAS value, as we also observed for structure 1 in Figure 5. Intersections among the individual experts are comparative rare. Furthermore, another point becomes apparent from the comparison of the individual colored lines between Figure 5 and Figure 7.

When comparing each individual colored line, the experts seem to repeat their role for the measurement of the SDAS for both different microstructures. They perform for most of the sub-images very similar measurements for both microstructures. This might corroborate the assumption that a manual SDAS measurement is somewhat influenced by the dendrite selection of the human expert. For the manual measurements made for the purposes of this work, a certain bias of the measured SDAS values is observed. Furthermore, this individual bias seems induced by the intuition of the individual human experts. The automated measured SDAS values from all sub-images lie within the standard deviations.

### 3.3. Computational Performance

The following section is written with the purpose of presenting the computational performance of the algorithm explained in Section 2 and to investigate the time requirements for the manual and the automated measurement procedure of the SDAS.

In Section 2.2, the procedure of the object segmentation is clarified. The microstructure from structure 1 consist of a relative high fraction of α-aluminium dendrite structure. Hence, the branches, consisting of several connected primary and secondary dendrites or fragments, are quite rarely interrupted by the solid solution. This results in complex and relatively big branches, which have to be processed by the algorithm. A contrary manner occurs for structure 2, with a relative low fraction of α-aluminium dendrites and a reasonably higher fraction of eutectic solid solution compared to structure 1. As a result, the objects, processed by the algorithm, are not as complex as those from structure 1. In Figure 8 we present the time requirements for the two different microstructures, using both measurement procedures. The filled bars in this chart illustrate the average values from the six manual measurements. Furthermore, the average values of the algorithm are represented by the dashed bars. Comparing structure 1 with structure 2, measured by means of the algorithm, it is apparent that not as much as time is required for structure 2 than for structure 1.

Hence, there seems to be a tendency, that there is an influence of the object size on the time requirement. As explained above, the size of the objects that are extracted from the binary image seems to influence the computational performance for the automated SDAS measurement. For the two microstructures investigated for the purpose of this work, the difference in time requirement between the manual and automated measurement procedures is more pronounced for structure 1 than for structure 2. Thus, we assess that there is a pronounced influence due to the fraction of the eutectic solid solution on the computational performance. The decreasing fraction of eutectic solid solution decreases the time required for it to be processed by the algorithm. Measuring the SDAS manually, the operator has to draw a line through the secondary dendrite arms, which is very time consuming. The assessment of the values in Figure 8, in consideration of the standard deviation bars, reveals that the automated measure procedure seems to be faster than a manual operator. The difference between a manual and an automated measurement of the SDAS is influenced depending on the fraction of eutectic solid solution of the microstructure. Independently of this time requirement advantage, the automated measure procedure requires less human resources compared to a manual measure procedure.

There are guidelines, that specify the measurement procedure of the SDAS, exemplary “P220” [16]. To the best of our knowledge, the measurement density of the SDAS has not been further specified or considered previously according to these guidelines. The automated measurement procedure explained in this work allows the measure density to be increased, with appropriate expense.

## 4. Conclusions

The following conclusions can be drawn from the present evaluation of the algorithm:The detected dendrite arms were grouped through Hough transformation. The individual groups were scored based on a set of scalar measurements that can easily be assigned to the individual groups. This allowed for the identification of individual secondary dendrite structures that are suitable for SDAS measurement.Evaluating the automated measurement procedure against the manual values of six operators, the results were in good agreement.The algorithms were tested against six sets of manual SDAS measurements for two different AlSi cast alloys. A good agreement between manual and automated measurements was found in all cases, indicating the effectiveness of the developed algorithms in measuring the SDAS.The algorithms always decided uniformly within its possibilities, which is why the measured value can be interpreted more uniformly.An analysis of the required computation times revealed the algorithm was on average faster at measuring the SDAS than a human operator. Thus, the developed algorithms allowed for an increase in the measurement density that is used to characterize the cast microstructures.An increased measurement density will decrease the selection bias of the results.

## Figures and Tables

**Figure 1 materials-14-02827-f001:**
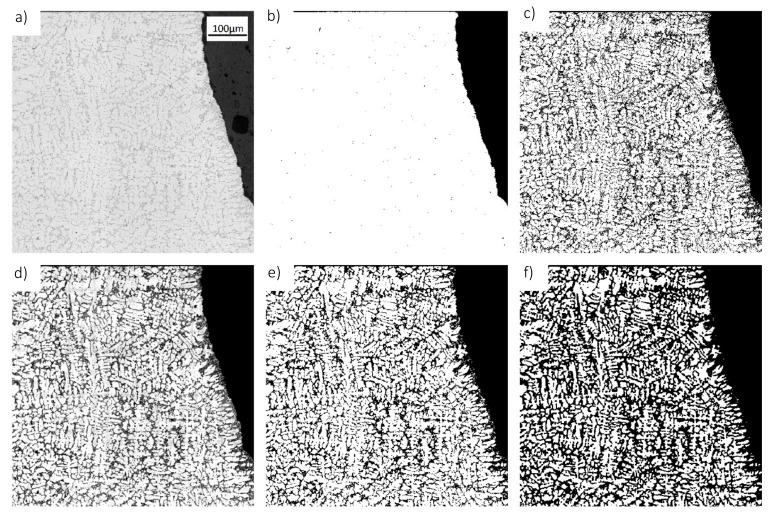
Illustration of the binarization procedure; (**a**) original grayscale image; (**b**) labeling of all alloy pixels through Otsu’s algorithm; (**c**) initial binary image obtained through Equation (1); (**d**) filtered image; (**e**) final binary image; (**f**) eroded binary image.

**Figure 2 materials-14-02827-f002:**
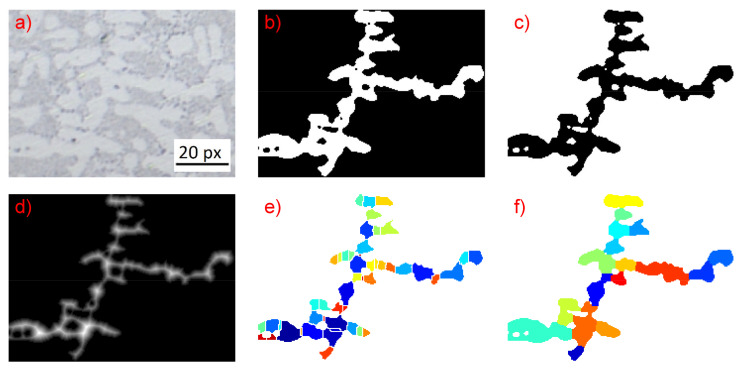
Procedure for decomposing of complex objects; (**a**) crop of the original micrograph showing a complex dendritic scheme. (**b**,**c**) Binarization of the dendritic scheme. (**d**) this approach shows a tendency for over-segmentation of the image. (**e**) In order to resolve this issue, a post-processing step is required that merges the individual watershed regions in such a way that the individual dendrite arms can be identified. To this end, neighborhood relations are defined for all watershed regions on the basis of a 4/8-connected neighborhood. (**f**) Afterwards, a procedure is initiated that aims at merging two neighboring watershed regions to a single one.

**Figure 3 materials-14-02827-f003:**
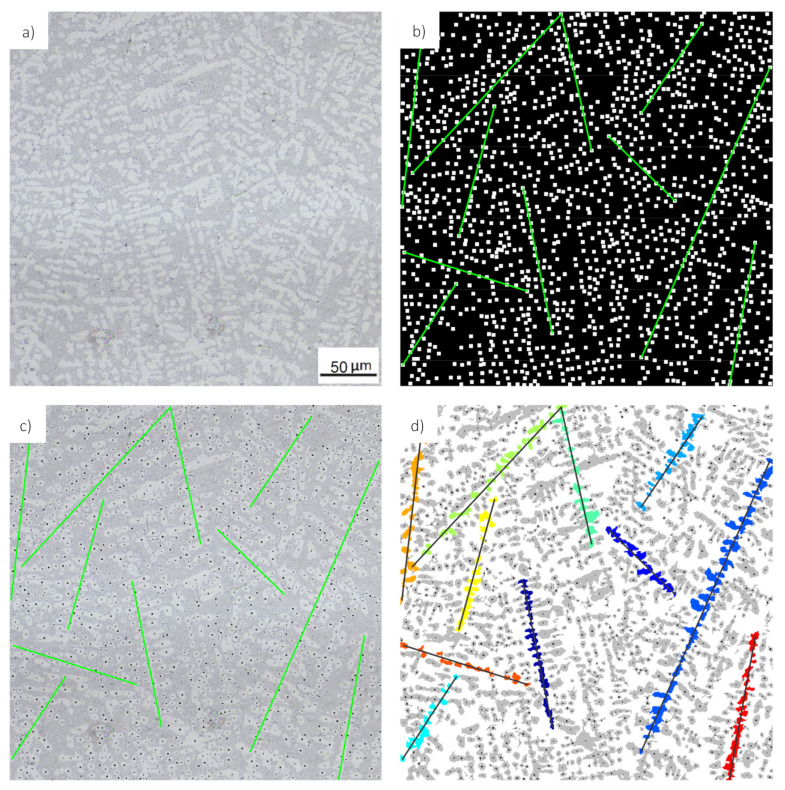
Segment clustering through Hough transform; (**a**) original micrograph; (**b**) dilated centroid mask (black/white) with a selection of detected lines (green); (**c**) line selection within the original micrograph together with the segment-centroids; (**d**) clustering of segments that intersect with a detected line.

**Figure 4 materials-14-02827-f004:**
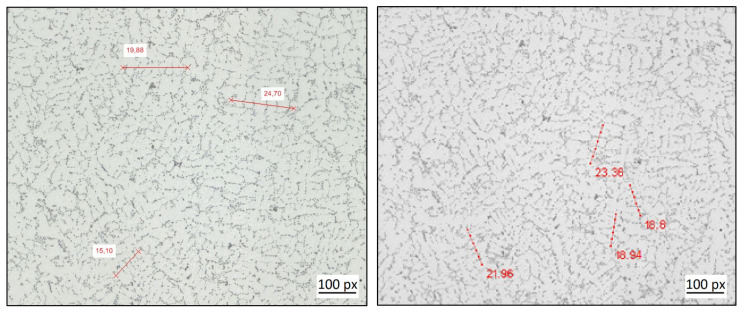
Comparison of manual (left) and automated (right) dendrite measurements exemplified for the 5th/6th sub-image of structure 1.

**Figure 5 materials-14-02827-f005:**
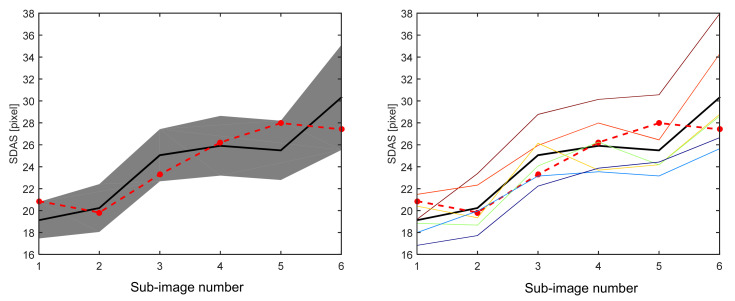
Comparison between the automated and manual SDAS measurements for the low eutectic structure; (left): comparison of the automated (red, dashed)–and manually measures (black, thick) SDAS values to the standard deviation (gray shaded region) of the distribution of manually measured SDAS values; (right): comparison of the automated (red, dashed)–and manually measures (black, thick) SDAS values to the individual manual SDAS measurements (thin, color).

**Figure 6 materials-14-02827-f006:**
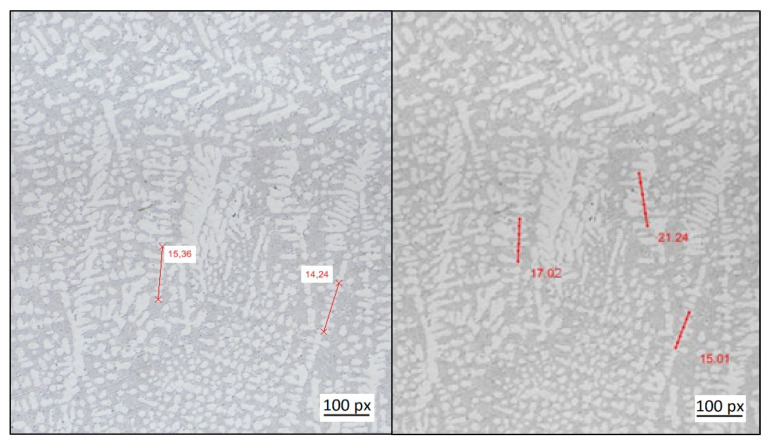
Comparison of manual (left)–and automated (right) dendrite measurements exemplified for structure 2.

**Figure 7 materials-14-02827-f007:**
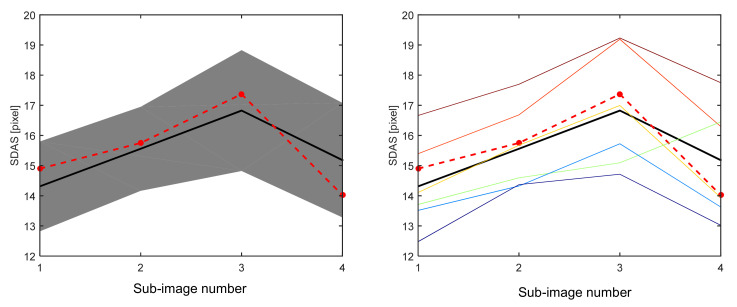
Comparison between the automated and manual SDAS measurements for the high eutectic cast; left: comparison of the automated (red, dashed) and manually measured (black, thick) SDAS values to the standard deviation (gray shaded region) of the distribution of manually measured SDAS values; right: comparison of the automated (red, dashed)–and manually measured (black, thick) SDAS values to the individual manual SDAS measurements (thin, color).

**Figure 8 materials-14-02827-f008:**
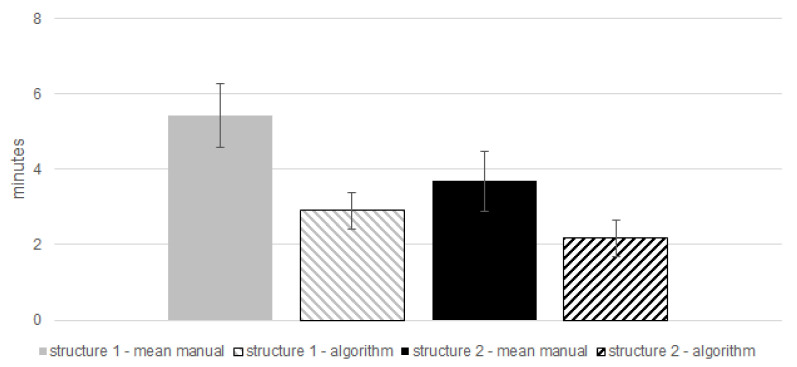
Mean time requirements per sub-image; mean manual–structure 1 (grey), algorithm–structure 1 (grey, dashed), mean manual–structure 2 (black), algorithm–structure 1 (black, dashed).

**Table 1 materials-14-02827-t001:** Parameter values that are employed within the individual processing steps of the algorithms.

Symbol	Value	Function
*N_sub_*	10 × 10	Number of sub-images into which the original image is divided
*T_1_*	0	Threshold value for initial binarization
*M_G_*	5 × 5	Size of the averaging mask for filtering of the binary image
*σ_G_*	5	Standard deviation of the Gaussian filter
*T_2_*	0.6	Threshold value for final binarization
*L_erode_*	3/0 *	Size of the structuring element (disk) used to erode the final binary image
*A_min_*	50	Minimal number of pixels of a valid object
*S_min_*	0.85	Minimal solidity value at which two neighboring segments are merged to a single one
*L_dilate_*	10	Size of the structuring element (square) used to dilate the centroid mask
Δ*θ*	1	Grid size of the angular coordinate used in the Hough transform
Δ*ρ*	1	Grid size of the radial coordinate used in the Hough transform
*N_peaks_*	3500	Number of identified Hough peaks
*H_min_*	0.1 * max(H)	Minimal intensity value at which a line is detected by the Hough transform
*L_min_*	50	Minimal length of a line detected via Hough transform
Δ*L*	100/40 **	Maximal gap width between two pixels that are associated by one line
*N_min_*	6	Minimal number of segments considered for cluster scoring
*N_max_*	10	Maximal number of segments considered for cluster scoring
*w_b_*	80	Weight penalizing non-homogeneous dendrite distance distributions
*w_c_*	10	Weight penalizing average SDAS that are above dmax
*w_d_*	15	Weight penalizing an average cluster orientation that is not perpendicular to the line orientation
*w_e_*	10	Weight penalizing an average cluster orientation that is not perpendicular to the line orientation
*w_f_*	−5	Factor weighting the average aspect ratio for cluster scoring
*μ_d,max_*	40/25 **	Maximal average dendrite arm spacing
*N_common_*	2	Maximal allowable number of common segments between two unique clusters

* In case of the AlSi11-alloy discussed in Section 3, the dendrite density within the image is sufficiently low, so that an erosion operation on the binary image is not necessary. The size of the structuring element was therefore set to zero, which is equivalent to not performing any erosion operation. ** The maximum value for the dendrite arm spacing as well as the maximal gap width between two pixels are dependent on the image resolution as well as on the size of the dendritic structures that are to be expected. A general definition of both values is not possible, and hence, both values have to be provided by the user.

**Table 2 materials-14-02827-t002:** Chemical composition (wt. %) of the cast alloys used in this work.

	Si	Fe	Cu	Mn	Mg	Ni	Zn	Pb	Ti	Al
AlSi10Mg(Cu) [EN AC-43200]	9–11	0.65	0.35	0.55	0.2–0.45	0.15	0.35	0.1	0.2	Balance
AlSi11 [EN AC-44000]	10–11.8	0.15	0.02	0.05	0.1–0.45	-	0.07	-	0.15	Balance

## Data Availability

The data used to support the findings of this study are available from the corresponding authors upon request.

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
