# Peer review of "Automatic Determination of Secondary Dendrite Arm Spacing in AlSi-Cast Microstructures"

_materials, 2021, doi:10.3390/ma14112827_

Round 1
Reviewer 1 Report
I have reviewed the submission and could not found any scientific novelty. Also, the presented method does not seem to be reliable enough as the authors did not discuss the possible errors and deviations from reality. In addition, the literature review is not sufficient and acceptable for the given topic. I do not think the paper could get enough priority to be considered for publication.
Author Response
Response to Reviewer 1 Comments
Point 1: Thank you for taking the time to read the article. We are very sorry that we could not sufficiently present the novelty content for you. However, the authors are not aware of any publication that describes a similar algorithm to determine dendrite arm spacing fully automatically. Semi-automated methods are often presented that require an indication of the radius of the dendrite tip or the approximate DAS. Furthermore, the object detection as well as the segmentation of the individual secondary arms (decomposing of complex objects) seems to be new in our opinion, for the application of DAS measurement with heterogeneous image contents. One of the difficult challenges is to identify the segmented objects as coherent meaningful structures.
Response 1: I have reviewed the submission and could not found any scientific novelty.
Point 2: The Figures 5 and 7 show the individual measurements of a total of 6 manual measurements performed by humans. This was done for 6 respectively 4 subimages. Each person decides differently when selecting and evaluating the SDAS measurement. The standard deviation of the manual measurements of the humans is shown in gray in the left image. For both of the structures investigated, the measurements of the algorithms are within the standard deviation of the manual measurements of the humans. In our opinion, the different selection of dendrites by different people is a particular problem, which often results in measurement uncertainties in practice. However, the algorithms always decide uniformly within its possibilities, which is why the measured value can be interpreted more uniformly.
|
Sub-image |
|
Sub-image |
Response 2: Also, the presented method does not seem to be reliable enough as the authors did not discuss the possible errors and deviations from reality.
We are very sorry that from your point of view we did not provide enough literature. Our concern with the number of literatures was that we focus on the relevance of SDAS measurement and the essential tools of digital image processing that we have used.
Response 3: In addition, the literature review is not sufficient and acceptable for the given topic. I do not think the paper could get enough priority to be considered for publication.
Reviewer 2 Report
- the following sentence, which is given in the Introduction, has also to be developed in more detail, especially with regard to the cited individual publications [1–5].
- add a magnification to the fig. 1, 4, 6,
- add scale to the fig. 2
- please extend the conclusion
Author Response
Thank you for taking the time to read the article. Our response can be found in the attachment.

Reviewer 3 Report
Dear Authors,
The paper is interesting and important for readers who analyze the microstructure of silicon-aluminum alloys, especially for refinement of α solid solution (DAS and SDAS).
However, the authors did not avoid some flaws which should be corrected and/or supplemented:
- p.1 line 27 - add information that modification of alloy also influences on α solid solution refinement. In this place it should also appear eutectic phase
- p. 3 Fig.1 illegible designations c) d) e) f)
- p.3 l. 99 add Figure 1a)
- p. 7 Fig3 illegible designations b). Subtitles should be Figure 3. not Figure 2.3
- p. 11 Figure 4 it does not make sense to indicate the value of DAS so precise 23.3634 um
- Fig. 5 and 7 add the names of X-axe
- first two conclusions are obvious.
Best regards
Author Response
Thank you for taking the time to read the article. You can find the answered comments in the attachment.

Reviewer 4 Report
Dear Authors,
Thank you for submitting the manuscript.
I have few general comments in addition to the ones highlighted in the pdf (attached along with this message).
1) Please include variables in terms of equations rather than inserting them as a regular text.
2) Would it be possible to share the algorithm of these measurements with the general audience?
3) How different is this algorithm from a commercial software like imageJ and Fiji?
4) Lastly, in automobile industry, the major emphasis is on the overall strength and microstructure. From that point of view, the performed work is a bit too much. However, from research (and solely research) purposes, the work is exceptional.
5) Please insert scales in the microstructures.
Thank you.

Author Response
Thank you for taking the time to read the article. Our response to the comments is attached.

Reviewer 5 Report
The papers describes an algorithm to automatically measure microstructural features and compares its performance to manual measurements, showing that the results are similar with a great time saving and, possibly, a lower selection bias.
I think that this paper is definitely worth publishing, but I would like that the authors could take care of some minor issues.
1. I have to admit that the expression "mix crystal" sounds quite unusual to me, and a quick research suggested to me that this expression is not commonly used for metals.
2. There is no indication about the software which was used. Was the routine written by the authors? Did they use a commercial software?
3. Please use the subscript. In most of the paper it is not indicated.
4. Though I wouldn't describe myself as an English expert, I would suggest to check some spelling (for example, at rows 324-326, there is an extra full point, a "significant" which I guess should be "significantly", "will described".
5. At Row 353 and 410: the authors refer to subimages which are not shown to the reader (or, at least, it is unclear to me).
6. Check the caption for Fig. 8. A legend within the plot could help the reader
7. At row 477 I guess there is a reference which is not properly addressed (“P220”)
Author Response
Thank you for taking the time to read the article. The response to the comments is attached.
